# Psychosocial family-level mediators in the intergenerational transmission of trauma: Protocol for a systematic review and meta-analysis

Emma J. Mew[1]*, Kate Nyhan[2,3], Jessica L. Bonumwezi[4], Vanessa Blas[5], Hannah Gorman[6], Rachel Hennein[7,8], Kevin Quach[5], Veronika Shabanova[9], Nicola L. Hawley[1], Sarah R. Lowe[6]

1 Department of Chronic Disease Epidemiology, Yale School of Public Health, Yale University, New Haven, Connecticut, United States of America, 2 Harvey Cushing/John Hay Whitney Medical Library, Yale University, New Haven, Connecticut, United States of America, 3 Department of Environmental Health Sciences, Yale School of Public Health, Yale University, New Haven, Connecticut, United States of America, 4 Department of Psychology, Montclair State University, Montclair, New Jersey, United States of America, 5 Yale College, Yale University, New Haven, Connecticut, United States of America, 6 Department of Social and Behavioral Sciences, Yale School of Public Health, Yale University, New Haven, Connecticut, United States of America, 7 Department of Epidemiology of Microbial Diseases, Yale School of Public Health, Yale University, New Haven, Connecticut, United States of America, 8 Yale School of Medicine, Yale University, New Haven, Connecticut, United States of America, 9 Department of Biostatistics, Yale School of Public Health, Yale University, New Haven, Connecticut, United States of America

* emma.mew@yale.edu

## Abstract

### Introduction

Family-level psychosocial factors appear to play a critical role in mediating the intergenerational transmission of trauma; however, no review article has quantitatively synthesized causal mechanisms across a diversity of trauma types. This study aims to systematically consolidate the epidemiological research on family-level psychosocial mediators and moderators to ultimately produce causal diagram(s) of the intergenerational transmission of trauma.

### Methods

We will identify epidemiological peer-reviewed publications, dissertations, and conference abstracts that measure the impact of at least one psychosocial family-level factor mediating or moderating the relationship between parental trauma exposure and a child mental health outcome. English, French, Kinyarwanda, and Spanish articles will be eligible. We will search MEDLINE, PsycINFO, PTSDpubs, Scopus, and ProQuest Dissertations and Theses and will conduct forward citation chaining of included documents. Two reviewers will perform screening independently. We will extract reported mediators, moderators, and relevant study characteristics for included studies. Findings will be presented using narrative syntheses, descriptive analyses, mediation meta-analyses, moderating meta-analyses, and causal

**Data Availability Statement:** Detailed project materials, resources, statistical code, and protocol

amendments are available on the Open Science Framework (https://osf.io/k5ezm/).

**Funding:** The author(s) received no specific funding for this work.

**Competing interests:** The authors have declared that no competing interests exist.

diagram(s), where possible. We will perform a risk of bias assessment and will assess for publication bias.

## Discussion

The development of evidence-based causal diagram(s) would provide more detailed understanding of the paths by which the psychological impacts of trauma can be transmitted intergenerationally at the family-level. This review could provide evidence to better support interventions that interrupt the cycle of intergenerational trauma.

## Trial registration

**Systematic review registration:** PROSPERO registration ID #CRD42021251053.

## Introduction

Intergenerational trauma is the process by which the psychological impact of a traumatic event is transmitted from one generation to the next [1–4]. This results in subsequent generations experiencing the psychological effects of a traumatic event without exposure to it, such as increased risk of developing trauma- and stressor-related disorders [1–4]. Epidemiological studies provide some evidence for intergenerational trauma. Many studies to date have focused on the descendants of Holocaust survivors [5], whereas smaller bodies of literature have included other populations and traumatic events [1–4, 6, 7], focusing primarily on the transmission of trauma-related symptoms from the first (G1) to the second (G2) generation.

Researchers have evaluated intergenerational trauma through two main paths: (1) the path from parental trauma exposure to child psychopathology; and (2) the path from parental trauma-related symptoms to child psychopathology. One recent meta-analysis showed that being a child of a Holocaust survivor was modestly associated with increased posttraumatic stress disorder (PTSD) symptoms [8] and another meta-analysis of studies across any population or trauma-type identified a significant association between parental PTSD symptoms and child PTSD symptoms [9]. However, evidence is mixed, as additional reviews have been less conclusive [10, 11]. Researchers also speculate transmission from G1 to the third (G3) generation, which is supported by animal models [12, 13]. However, few epidemiological studies have evaluated transmission from G1 to G3 and the few that have obtained mixed results [14].

In addition to empirically demonstrating intergenerational trauma, it would be helpful to identify the causal processes underpinning this phenomenon as such findings could inform interventions aimed to prevent or mitigate it. Mediation studies examine the extent to which an intermediate variable (mediator) explains the effect of exposure on an outcome in an attempt to tease apart causal mechanisms [15]. Mediation studies quantify both direct and indirect effects, meaning the effect of exposure on the outcome and the effect of the exposure on the outcome through a mediating variable [15]. Epidemiological studies can also identify potential moderators, or variables that influence the degree of association between an exposure and an outcome. These relationships are often demonstrated visually through causal diagrams, most formally directed acyclic graphs (DAGs). DAGs can powerfully deduce causal relationships as they incorporate statistical approaches into a visual diagram [16]. Fig 1 demonstrates simplified DAGs for the intergenerational transmission of trauma including mediation (Fig 1A) and moderation (Fig 1B) paths.

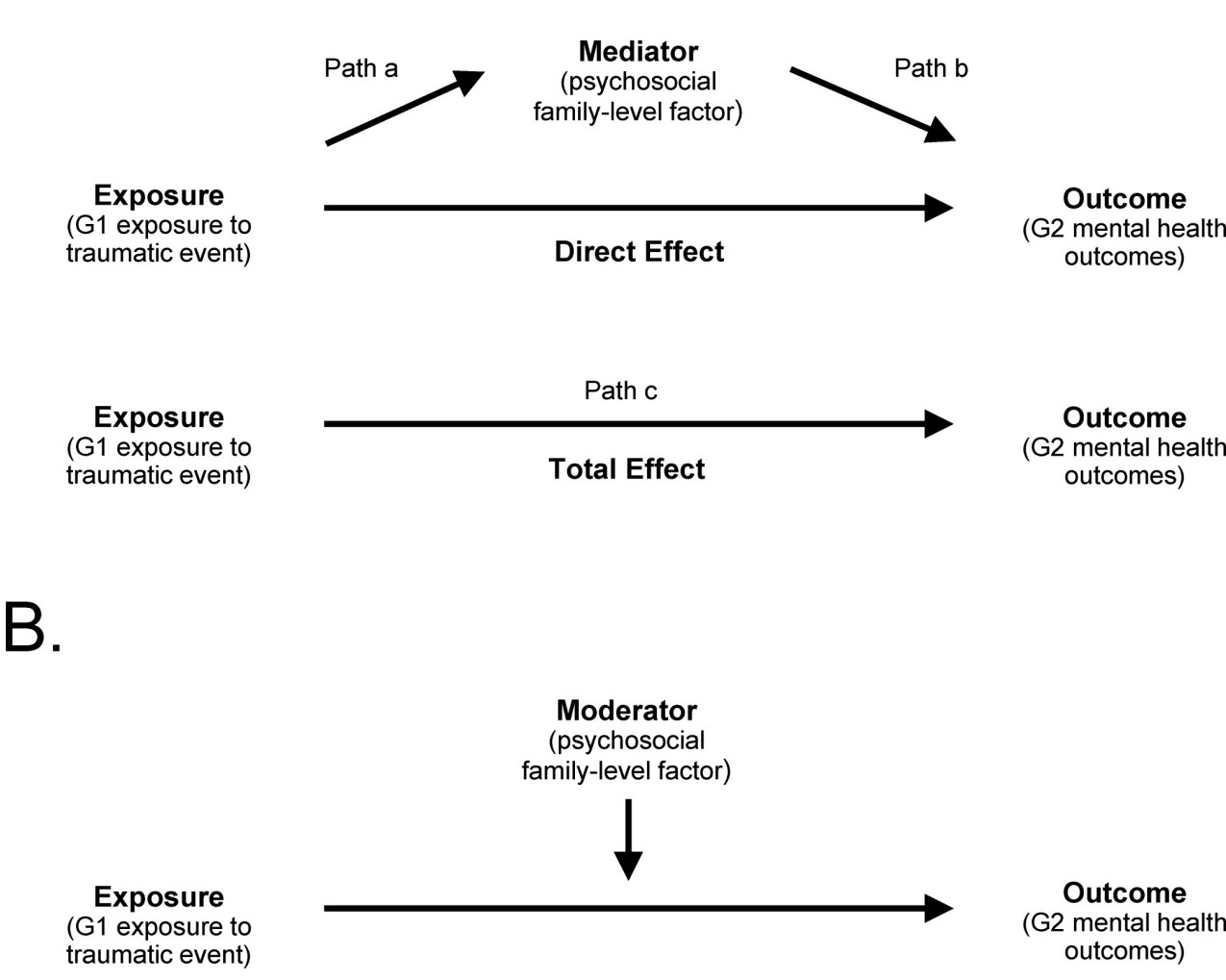

**Fig 1. Simplified directed acyclic graphs for the intergenerational transmission of trauma from first (G1) to second (G2) generation through parental trauma exposure F0E0 child mental health outcome.** (A) Single mediator model showing indirect, direct, and total effects. (B) Simple moderation model.

Evidence suggests several complex, multifaceted causal processes underpin the intergenerational transmission of trauma. Although the most notable evidence is for genetic and epigenetic mechanisms, these do not fully account for transmission of trauma, suggesting the involvement of other psychosocial and interpersonal factors [17, 18]. Recent studies have provided some evidence that psychosocial factors that operate within the household play some role in mediating the intergenerational transmission of trauma [1, 3, 4, 17], with the strongest evidence supporting parenting behaviors, parental mental health symptoms, and attachment [1, 3, 4].

Understanding the role of psychosocial factors within the family in mediating and moderating the intergenerational transmission of trauma is critical to develop household-level psychosocial interventions that prevent transmission among high-risk families. Some psychosocial interventions have been explicitly developed to prevent the intergenerational transmission of trauma or have been evaluated for this purpose [19, 20]; however, these interventions are

heterogeneous in their theoretical and therapeutic approaches and in the evidence used to their inform development. Understanding mediating and moderating factors could also optimize existing interventions, helping researchers understand what intervention component(s) are most effective and in what context(s).

To assess the need for a systematic review on this topic [21], we conducted a preliminary search to identify relevant literature reviews (eTable1 in S1 File). We located 21 reviews, most of which were scoping reviews focused on narrow sub-populations and trauma-event types (eTable 2 in S1 File). Only two reviews conducted moderation meta-analyses: one restricted to mothers with childhood maltreatment, and the other only including studies published until 2011. Based on these findings: (1) no review has published a diagram to demonstrate and contextualize epidemiological causal relationships at the family-level in the intergenerational transmission of trauma; (2) no review has meta-synthesized family-level mediators; and (3) there is a need to perform an updated meta-analysis of family-level moderators across diverse trauma-types.

We aim to fill these gaps by systematically consolidating the family-level psychosocial mediators, their respective direct and indirect effects, and moderators of such effects, on the intergenerational transmission of trauma. We will focus on a broad scope of populations and trauma types to increase the sample size available for meta-synthesis and to understand general mechanisms. We will also include impacts on the third generation (G3). By using the consolidation and meta-synthesis of epidemiological data, we will generate the first causal diagram to describe mechanisms through which trauma is transmitted intergenerationally.

## Review questions

1. What family-level psychosocial mediator(s) and family-level psychosocial and demographic moderator(s) have epidemiological evidence supporting their involvement in the intergenerational transmission of trauma?

2. What are the combined quantitative effects across mediators and moderators in the intergenerational transmission of trauma?

## Materials and methods

This project will follow review guidelines from the Joanna Briggs Institute (JBI) Manual for Evidence Synthesis [21] and the Cochrane Handbook for Systematic Reviews [22]. We followed the PRISMA-P [23] (eTable 3 in S1 File) reporting guideline to develop this protocol. This study has been registered on PROSPERO (registration ID #251053). We will post project materials, statistical code, and protocol amendments on our Open Science Framework webpage (https://osf.io/k5ezm/).

### Inclusion criteria

We will use the systematic review PICOTS (Population, Intervention, Comparator, Outcome, Timeframe, Study design) framework to define eligibility, adapted to accommodate our research question focused on observational studies reporting meditator(s) and/or moderator (s) (Table 1). Definitions/elaborations documents will be available on Open Science Framework (https://osf.io/k5ezm/).

**Population and exposure(s).** Included studies will have a population composed of at least one primary caregiver (G1) who experienced a traumatic event and their child (G2) and/or grandchild (G3) who did not experience this traumatic event firsthand. The G1 individual

**Table 1. Eligibility criteria for systematic review.**

| Inclusion concept | Eligibility criteria |
| --- | --- |
| Population | • Human population |
| Exposure | • At least one primary caregiver (G1) who experienced a traumatic event and their child (G2) and/or grandchild (G3) who did not experience this traumatic event<br>• The traumatic event in G1 must have occurred before or during the birth of the child under study |
| Mediator or moderator | • Psychosocial factor(s) operating at the family-level that influence the interpersonal relationship between the child and their immediate family and/or primary caregiver before the child has reached 18 years of age. Factors can be a mediator or moderator<br>• Comparators will be defined as the absence, or varying levels, of factor(s) measured in at least some sub-group of participants |
| Outcome | • Quantitative measurement of at least one total, direct, or indirect effect of trauma (psychological symptoms/functioning) in the child under observation at any point during the child's lifespan (including >18 years old) |
| Timeframe and language | • No date restrictions, but limited to English, French, Kinyarwanda, and Spanish languages (research team members are fluent in these languages) |
| Study design and document type | • Original research article of a quantitative or mixed methods study published in journals, dissertations (any level), or conference abstracts |

would not need to directly report their traumatic exposure, as long as there is some method to assess this, such as indirect reporting by G2. The traumatic event in G1 must have occurred before the birth of G2. G1 or G2 parents do not need to be biologically related to the child of interest. We will include trauma-exposed parents regardless of their trauma-related symptomatology. Animal studies will be excluded.

A traumatic event will be defined as exposure to threatened death, serious injury, or sexual violence, as outlined in the Diagnostic and Statistical Manual of Mental Disorders [24]. We will also consider parental trauma-related symptoms as the exposure, contingent that these symptoms are a direct result of an eligible traumatic exposure.

**Mediator(s) and moderator(s).**   Included studies will have at least one psychosocial factor operating at the family-level (including household- and parent-levels) that measured the interpersonal relationship between the child and their immediate family and/or primary caregiver before the child has reached 18 years of age. These could be framed or examined as mediators or moderators. Original articles would not need to explicitly use these terms or conceptualize factors in this way to be eligible.

Psychosocial factors might include parental psychiatric symptoms, suggesting that parental trauma-related mental health symptoms could be considered as the exposure or the mechanism, depending on the context. Factors might also include parental psychological factors, interpersonal parent-child relationships, or social factors within the household-level (which include demographic factors) that would influence child-rearing behaviors [25]. Comparators will be defined the measurement of the absence of the factor(s) or varying levels of continuously-measured factor(s) of interest in at least some sub-group of participants in the study sample. We will also include demographic factors (for example, age, gender, socioeconomic status, etc.) as moderators.

**Outcome(s).**   Included studies will quantify either the presence or absence of direct or indirect effects of trauma in the child (G2 or G3) under observation. Given the heterogeneity in clinical presentations of intergenerationally-transmitted trauma, we will only include quantitatively measured markers of psychological symptoms/functioning (i.e., child psychological development, psychopathology, and/or indicators of wellbeing). To increase our yield, we will be inclusive in what outcomes would qualify as child psychopathology and/or indicators of wellbeing for screening and may restrict this criterion after we assess the final sample of

included studies. Outcomes will be considered in offspring. Psychiatric symptoms in the child would not need to be directly observed; for example, it would be acceptable for G1 to report on G2 symptoms.

**Date and language restrictions.** This review will have no date restrictions, but will be limited to studies published in English, French, Kinyarwanda, and Spanish languages, for feasibility, as research team members are fluent in these languages.

**Study design and document type.** We will include any epidemiologic study design, such as cross-sectional studies, case-control studies, cohort studies, randomized controlled trials, or the quantitative component of mixed-methods studies. Systematic review papers, meta-analyses, case-studies, qualitative studies, and study protocols will be excluded.

Due to feasibility, we will only include published literature (peer-reviewed literature and dissertations). One exception is that we will include conference abstracts during the screening phase contingent upon gaining access to the full text document (see Source of evidence selection). We included conference abstracts and dissertations to reduce publication bias.

## Searches

The search strategy was developed and refined by a medical research librarian (KN) with the assistance of the first author. The final search strategy was developed using the following Boolean logic: *[intergenerational concept] AND [family level concept] AND [transmission of trauma concept]* using appropriate keywords and controlled vocabulary.

We will search MEDLINE, PsycINFO, PTSDpubs (formerly PILOTS), Scopus (conference papers only), and ProQuest Dissertations and Theses (doctoral dissertations only). eTables 4–8 in S1 File shows the final search strategy for each bibliographic database. Although we will only retrieve doctoral dissertations from Scopus, our PsycINFO search will capture dissertations of all levels. The final search was peer-reviewed by a second medical librarian using the Peer Review of Electronic Search Strategies (PRESS) checklist before the search was finalized [26].

We limited the search to five databases for feasibility given large preliminary search yields. However, to compensate for this limitation, we will implement three methods to increase sensitivity: (1) robust forward citation chaining of included studies; (2) reference list screening of relevant reviews from the authors' personal libraries; and (3) screening additional papers from the authors' personal libraries.

## Source of evidence selection

Title/abstract and full-text screening will be performed independently by two reviewers using Covidence software [27]. Discrepancies will be resolved through screening from a third reviewer. During full-text screening, we will record the primary reason for exclusion.

We will screen the full-text versions of conference abstracts during full-text review. For each conference abstract, we will first attempt to locate full-text document online or by emailing the corresponding author. If we cannot obtain the full text using this process, we will exclude the record.

## Data extraction

We will use Research Electronic Data Capture (REDCap) software to extract data from included studies [28]. Data extraction will be performed by one reviewer in combination with quality assurance checks. Discrepancies will be resolved through consensus, and when needed, discussion with a third reviewer. We will de-duplicate the results of individual studies reporting on the same underlying project and will use this combined information for our analyses.

eTable 9 in S1 File shows the drafted data extraction items, which will likely be refined after piloting.

## Training and piloting procedures

We will hold piloting and training procedures for reviewers before each screening stage. To prepare for title/abstract screening, reviewers will first review training materials, which will include a draft definitions/elaborations document. We will then undergo a series of training rounds to assess inter-rater agreement and identify discrepancies. In each round, reviewers will independently screen a random sample of 25 records from our final search yield to calculate percent agreement. Then, after each round, the team will discuss discrepancies and modify the definitions/elaborations document. We will repeat this process until we achieve team agreement of >75%, and will then begin title/abstract screening. We will follow similar procedures for full-text review training. Reviewers will first review training materials and draft definitions/elaborations document for full-text screening. We will then undergo a series of training rounds where each reviewer will independently screen a sample of approximately 10 records. Each round will be followed by discussion to modify screening procedures. We will begin full-text screening after we achieved agreement of >75%.

Prior to beginning data extraction, the first author will prepare and disseminate training materials for data extractors. Then, the team of data extractors will pilot data extraction and critical appraisal of 1–2 papers, discuss discrepancies, and repeat the process until achieving an agreement of >75% on each item before beginning extraction and appraisal.

## Data charting

We will present summary information for all included studies in tabular format (eTable 10 in S1 File). We will then group the mediators presented in eTable 10 in S1 File together into similar thematic concepts, through discussion between the first and senior author. We will present these findings in a second table (eTable 11 in S1 File).

## Meta-analyses

Our statistical analysis plan is presented below with the understanding that our analytical strategies may be refined based on the distribution of data collected and from input of statistical experts. Not all included studies will be eligible for meta-analysis. The first author in consultation from the senior author and project biostatistician (VS) will use information on the quality of reporting of results to decide what included studies will be eligible for meta-analyses. We will conduct meta-analyses using R statistical software and will consider a p-value < 0.05 as sufficient evidence to reject a null hypothesis.

**Mediation meta-analyses.** We will use a mediation meta-analysis approach to pool data from individual mediation studies that report enough statistical information to derive indirect, direct, and/or total effects at the path-level. This could include reporting of beta coefficients directly (standardized or not standardized), or through back-calculating these estimates using correlation matrices, means, and variances/standard deviations for the exposure or traumatic exposure/symptoms (x), mediator of interest (m), and outcome or mental health status (y) variables. Our primary analyses will assess the following pathway: parental trauma exposure → psychosocial mechanisms (including parental trauma-related symptoms) → child mental health outcome. We will also assess the following pathway: parental trauma-related symptoms → psychosocial mechanisms → child mental health outcome. We will conduct several meta-analyses according to each mediating construct and whether it mediated G1 to G2, G2 to G3, or G1 to G3 effects. We will conduct sensitivity analyses to test whether we could consolidate

findings of individual studies with different study designs and consolidate individual studies that measured the exposure as parental trauma versus parental post-traumatic symptoms. We plan to merge outcomes together to assess the global effect on child psychopathology; if there is enough data, we will sub-divide outcomes into different meta-analytic models. We will conduct meta-analyses following guidelines reported in the literature [29–31] that would be most suitable to for the methods reported in our sample of included studies, depending on the distributions and availability of statistical information reported in the included studies. We will aim to pool data as long as there are at least two beta estimates available for an eligible path.

**Moderation meta-analyses.**   We will aim to use a moderation meta-analysis approach to pool moderator data from individual studies, using similar methods as described above. Our primary analyses will assess moderating effects on parental trauma exposure → child mental health outcome; our secondary analyses will assess moderating effects on parental trauma-related symptoms → child mental health outcome. We will perform moderation meta-analyses using standard statistical methods [32], though we will take an atypical analytic approach given the nature of our research question. Traditional moderation meta-analyses assess the impact of study-level factors on the degree of association between an exposure and outcome [33, 34] and are restricted to assessing moderators that apply to every participant within analyses. For example, this could include moderating factors such as study design (such as whether cross-sectional studies versus cohort studies moderate the exposure-outcome relationship) or other population-level characteristics (such as study location or public versus private funding source). For the purpose of this review, however, it is unlikely this meta-analytic approach would be feasible to understand the moderating role of psychosocial family-level factors on the intergenerational transmission of trauma, as it is unlikely that individual studies would restrict their analyses to different levels for these types of factors, and thus, it would be unlikely that we would retrieve enough studies to assess differences in effect sizes using this traditional approach. For example, if we aimed to understand the moderating effect of parental communication style, it is unlikely that individual studies assessing the relationship between parental trauma exposure and child mental health outcome would have restricted their analyses to one specific level of communication style. To increase our likelihood of pooling estimates, and provide more meaningful narrative syntheses, we instead plan to combine individual studies that directly assessed for moderation as part of their analyses, as reported elsewhere [35]. We would aim to pool this data using an integrative data analysis approach (otherwise known as individual participant data meta-analysis) and would assess moderation using joint tests of interaction using linear regression models [33], though this will be contingent on securing original datasets from the included studies. Should it be challenging to obtain the individual-level datasets, we will report moderation findings using narrative syntheses.

**Assessment of methodological quality.**   We will assess the quality of mediation paths using the quality assessment tool used in other mediation meta-analyses [15]. To our knowledge, this is the only tool available to evaluate the quality of mediation studies. To assess the quality of moderation paths, we will assess the quality of included articles using the JBI critical appraisal tools, selected based on the study design of the included paper [36, 37]. We will apply this framework at the study level. We selected this tool as, to our knowledge, there are no tool developed specifically to evaluate the quality of moderation analyses. If we gather enough studies, we will conduct sensitivity analyses to assess whether study methodological quality impacts pooled estimates.

## Publication bias

We aim to assess for publication bias using visual funnel plot inspection for each mediation meta-analysis conducted for each mediating construct. We do not plan to develop funnel plots

for each moderating construct, as we do not anticipate having enough statistical information given that we anticipate only a small sample of included studies for each construct.

### Presentation of results

Though we will aim to conduct meta-analyses wherever possible, in instances where this would not be possible (for example, not enough statistical information was reported, but the study provides quantitative information that would be qualitatively relevant to understanding causal pathways), we will report this information using narrative syntheses. We will also develop at least one causal diagram using data generated from pooled effect estimates. These diagram(s) will visually demonstrate the consolidated quantified relationships between variables and their relative magnitudes of association. We will include paths that are explicitly supported in our data and theoretical paths not explicitly supported (differentiated using solid versus dashed path lines). We will construct DAGs where possible if we obtain sufficient information, which would include combined effect estimates from our meta-analyses.

### Assessing confidence in cumulative evidence

We will interpret our results using frameworks for causal inference [38] and/or Grading of Recommendations, Assessment, Development and Evaluations (GRADE) [39] to provide a narrative summary and recommendations for interventions.

## Discussion

### Strengths and limitations

Our methods are rigorous and follow the Cochrane Handbook for Systematic Reviews and Joanna Briggs Institute systematic review methods manual. We will take additional precautions to reduce likelihood of publication bias through forward citation chaining and including conference abstracts and dissertations. This review has several potential limitations. Although we consider our search strategy comprehensive, we will only search five databases and will not include Embase. We will not cover all languages, which might contribute language bias. We also might not locate enough studies to perform meta-analyses, and would instead produce narrative summaries of quantitative results. Our causal diagrams(s) will be restricted to psychosocial household-level factors and will exclude other relevant mechanisms outside of this scope such as biological factors. This review will also be unable to assess complex causal relationships between mediators, moderators, and their respective interactions; however, we will include this information in our narrative synthesis and causal diagram(s) if this information happens to be reported in individual studies.

## Conclusions

The development of evidence-based causal diagram(s) would provide more detailed understanding of the paths by which the psychological impacts of trauma can be transmitted intergenerationally at the family level, including the relative strength of each factor in mediating and moderating cause and effect. These results could then be applied to design and optimize evidence-based interventions that target mechanisms with the strongest mediating effect. Taken together, this review could provide evidence to better support interventions that interrupt the cycle of intergenerational trauma for future generations.

## Supporting information

**S1 Checklist.**
(DOCX)

**S1 File.**
(PDF)

## Acknowledgments

We thank Rhayna Poulin for administrative support.

## Author Contributions

**Conceptualization:** Emma J. Mew, Jessica L. Bonumwezi, Sarah R. Lowe.

**Data curation:** Kate Nyhan.

**Formal analysis:** Veronika Shabanova.

**Investigation:** Emma J. Mew.

**Methodology:** Emma J. Mew, Kate Nyhan, Jessica L. Bonumwezi, Veronika Shabanova, Nicola L. Hawley, Sarah R. Lowe.

**Project administration:** Emma J. Mew, Kate Nyhan, Vanessa Blas, Hannah Gorman, Rachel Hennein, Kevin Quach, Sarah R. Lowe.

**Resources:** Kate Nyhan.

**Software:** Veronika Shabanova.

**Validation:** Emma J. Mew, Sarah R. Lowe.

**Writing – original draft:** Emma J. Mew.

**Writing – review & editing:** Emma J. Mew, Kate Nyhan, Jessica L. Bonumwezi, Vanessa Blas, Hannah Gorman, Rachel Hennein, Kevin Quach, Veronika Shabanova, Nicola L. Hawley, Sarah R. Lowe.

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
