## [Decision Letter · Decision Letter 0]

17 Aug 2022

PONE-D-22-17606Psychosocial family-level mediators in the intergenerational transmission of trauma: Protocol for a systematic review and meta-analysisPLOS ONE

Dear Dr. Mew,

Thank you for submitting your manuscript to PLOS ONE. After careful consideration, we feel that it has merit but does not fully meet PLOS ONE’s publication criteria as it currently stands. Therefore, we invite you to submit a revised version of the manuscript that addresses the points raised during the review process.

We look forward to receiving your revised manuscript.

Kind regards,

Michael McCaul, MSc, PhD

Academic Editor

PLOS ONE

Journal Requirements:

3. We note that this manuscript is a systematic review or meta-analysis; our author guidelines therefore require that you use PRISMA guidance to help improve reporting quality of this type of study. Please upload copies of the completed PRISMA checklist as Supporting Information with a file name “PRISMA checklist”.

Reviewers' comments:

Reviewer's Responses to Questions

**Comments to the Author**

1. Does the manuscript provide a valid rationale for the proposed study, with clearly identified and justified research questions?

Reviewer #1: Yes

Reviewer #2: Yes

Reviewer #3: Yes

2. Is the protocol technically sound and planned in a manner that will lead to a meaningful outcome and allow testing the stated hypotheses?

Reviewer #1: Yes

Reviewer #2: Partly

Reviewer #3: Yes

3. Is the methodology feasible and described in sufficient detail to allow the work to be replicable?

Reviewer #1: Yes

Reviewer #2: No

Reviewer #3: Yes

4. Have the authors described where all data underlying the findings will be made available when the study is complete?

Reviewer #1: Yes

Reviewer #2: No

Reviewer #3: Yes

5. Is the manuscript presented in an intelligible fashion and written in standard English?

Reviewer #1: Yes

Reviewer #2: Yes

Reviewer #3: Yes

6. Review Comments to the Author

You may also provide optional suggestions and comments to authors that they might find helpful in planning their study.

Reviewer #1: Title: Psychosocial family-level mediators in the intergenerational transmission of trauma: Protocol for a systematic review and meta-analysis

Goals:

• Meta-analyses of epidemiological research of causal mechanisms across a diversity of family level mediators and moderators that play a role in the intergenerational transmission across diverse trauma-types.

Strengths

• This is a very comprehensive proposal for a systematic review and meta-analysis that would be a much-needed contribution to the literature.

• Study proposes to use well-established rigorous methodology/ framework: Cochrane Handbook for Systematic Reviews and Joanna Briggs Institute scoping review methods manual and addressing publication bias through forward citation chaining and including conference abstracts and dissertations.

• Inclusion of a broad range of psychosocial factors that might be mediators or moderators (e.g., trauma related mental health functioning, other psychological factors, interpersonal factors or social factors) between G1 exposure and G2 or G3 outcomes.

• Study will produce directed acyclic graph i.e. causal diagram(s) of identified mediators and moderators of intergenerational transmission of trauma.

Minor suggestions for improvement:

The term "unresolved trauma" in the discussion section of the abstract and again towards the end of the comes as a surprise and appears unrelated to the Introduction section where the Phrase "intergenerational transmission of trauma" and its familial mediators are noted. In fact this term is not used in the rest of the manuscript until the conclusion. Rephrasing to be consistent across sections or defining the term clearly at the outset would be helpful.

Reviewer #2: The investigators present a protocol for a systematic review to identify factors associated with intergenerational transmission of trauma.

The background is well written and covers the literature on the topic, the knowledge gap, and the need for this systematic review.

The methodology is sound and justified.

I have the following comments.

1. Duplicate data extraction should not be based on feasibility, but a matter of principle to reduce errors. Single extraction and quality control is also acceptable.

2. Dissertations are not generally considered to be peer-reviewed and may not be an appropriate source of information.

3. It is an excellent idea to consult experts regarding the statistical analysis plan. It is even better to use this advise to develop the protocol. While there are many unknowns, it would be helpful to present some decision rules in this regard. For example, how much data do you need to decide about statistical pooling or not. Two statements highlight this issue:

“We will conduct meta-analyses following guidelines reported in the literature (29) that would be most suitable to the methods reported in our sample of included studies” and “We will perform moderation meta-analyses using standard statistical methods. (30)”

Moderation meta-analyses are not standard and should be described in more detail.

This is also apparent in the abstract: “Findings will be presented using narrative syntheses, descriptive analyses, mediation meta-analyses, moderating metaanalyses, and causal diagram(s), where possible.” There is no indication of how you will decide which approach to use. These decisions should not be made after seeing the data.

4. Another issue is the evaluation of study quality. The authors state that they will use the tools applied in other mediation meta-analyses. This is not appropriate. The tool must be stated and described, with details on the items assessed and how the information is interpreted.

5. It would not be appropriate to use separate tools to assess study quality if this is not linked to the purpose of the SR. For example, an RCT may have high methodological quality (low risk of bias) but have other flaws that make it a poor study for evaluating moderation. The QUIPS tool can be used to assess risk of bias in prognosis studies, irrespective of the study design. If there is an equivalent tool for mediation studies, it should be the primary approach for ascertaining the study quality.

6. Please provide details on how you will create a funnel plot to assess publication bias given the nature of your outcomes and data. Will you create one for each moderator/mediator?

7. There is no information on Assessing confidence in cumulative evidence. The sentence under this section is not related to confidence in cumulative evidence. GRADE can be used for a variety of evidence types including narrative synthesis (GRADE CERQAL).

Overall, the rational for this project is sound, but there is insufficient detail for the work to be implemented and too many decisions will be made based on the nature of the data. This is an important concern.

Reviewer #3: My comments are about the search methods. The authors conducted a simple preliminary search to identify previously published reviews on the topic, in order to avoid duplication of effort and refine their topic. This search is well reported in the Supplementary materials. However, they also already performed what they refer to as the final search, in January and March 2021, in selected databases, with the assistance of a medical librarian; this search was peer-reviewed by a second medical librarian using the PRESS checklist. The search terms for each database are reported in the Supplementary materials and they look appropriate and well selected. Also, the reasons for using search limits (languages) are reported and explained.

I assume that the "full search" was run in order to select the appropriate search terms for each different database and show them in the protocol, but my concern is that the full search should not have been already run and reported in the protocol. By the time the protocol is published the search will need to be repeated to bridge the gap in the results, and for the general reader it is confusing to read of the search in the past tense when all the rest of the protocol is in the future tense. It is not necessary to see the number of results for each search line when they will be different once the search is re-run. Another reason not to perform the searches before the protocol is accepted for publication is that it is always possible that referees will suggest some changes to the search terms, or even that in the meantime there will be changes to the controlled vocabulary (such as Mesh) terms.

I suggest to remove mentions of the full search in this protocol, report the search strategies for each database without the number of results for each line, and run the full searches once the protocol is published.

7. PLOS authors have the option to publish the peer review history of their article (what does this mean?). If published, this will include your full peer review and any attached files.

Reviewer #1: **Yes: **Archana Basu

Reviewer #2: **Yes: **Lawrence Mbuagbaw

Reviewer #3: **Yes: **Dr Vittoria Lutje, Information Specialist, Cochrane Infectious Diseases group

---

## [Author Response · Author response to Decision Letter 0]

1 Oct 2022

Please see table in Response to Reviewers document.

---

## [Editor Report · Decision Letter 1]

13 Oct 2022

Psychosocial family-level mediators in the intergenerational transmission of trauma: Protocol for a systematic review and meta-analysis

PONE-D-22-17606R1

Dear Dr. Mew,

We’re pleased to inform you that your manuscript has been judged scientifically suitable for publication and will be formally accepted for publication once it meets all outstanding technical requirements.

Kind regards,

Michael McCaul, MSc, PhD

Academic Editor

PLOS ONE

---

## [Editor Report · Acceptance letter]

21 Oct 2022

PONE-D-22-17606R1 

Psychosocial family-level mediators in the intergenerational transmission of trauma:
Protocol for a systematic review and meta-analysis 

Dear Dr. Mew:

I'm pleased to inform you that your manuscript has been deemed suitable for publication in PLOS ONE. Congratulations! Your manuscript is now with our production department. 

Kind regards, 

on behalf of

Dr. Michael Gilbert McCaul 

Academic Editor

PLOS ONE